# Nutrition Education Programs for Adults with Neurological Diseases Are Lacking: A Scoping Review

**DOI:** 10.3390/nu14081577

**Published:** 2022-04-10

**Authors:** Rebecca D. Russell, Lucinda J. Black, Andrea Begley

**Affiliations:** 1Curtin School of Population Health, Curtin University, Perth, WA 6102, Australia; rebecca.d.robinson@postgrad.curtin.edu.au (R.D.R.); lucinda.black@curtin.edu.au (L.J.B.); 2Curtin Health Innovation Research Institute, Curtin University, Perth, WA 6102, Australia

**Keywords:** behavior change techniques, behavior change theories, dietary guidelines, neurological diseases, nutrition education

## Abstract

The nutrition recommendation for most common neurological diseases is to follow national dietary guidelines. This is to mitigate malnutrition, reduce the risk of diet-related diseases, and to help manage some common symptoms, including constipation. Nutrition education programs can support people in adhering to guidelines; hence the aim of this scoping review was to explore what programs have been implemented for adults with neurological diseases. We conducted this review according to a published *a priori* protocol. From 2555 articles screened, 13 were included (dementia n = 6; multiple sclerosis n = 4; stroke survivors n = 2; Parkinson’s n = 1). There were no programs for epilepsy, Huntington’s, and motor neurone disease. Program duration and number of sessions varied widely; however, weekly delivery was most common. Just over half were delivered by dietitians. Most did not report using a behavior change theory. Commonly used behavior change techniques were *instruction on how to perform a behavior*, *credible source*, and *behavioral practice/rehearsal*. Evidence of nutrition education programs for adults with neurological diseases is lacking. Of those that are published, many do not meet best practice principles for nutrition education regarding delivery, educator characteristics, and evaluation. More programs aligning with best practice principles are needed to assess characteristics that lead to behavior change.

## 1. Introduction

Neurological diseases are an increasing cause of morbidity and mortality and they are now the leading cause of disability and the second leading causing of death, globally [1,2]. Common neurological diseases include Alzheimer’s disease and other dementias, epilepsy, Huntington’s disease, motor neurone disease, multiple sclerosis, Parkinson’s disease, and stroke [1,3,4], and they affect up to one billion people worldwide [2]. The nutrition recommendation for most of the common neurological diseases is to follow national dietary guidelines. Such guidelines vary between countries in the specific details, but overall they promote consumption of a wide range of nutritious foods from each of the defined food groups, and to limit consumption of highly processed foods and drinks that are high in added sugars, salt, saturated fat, and alcohol [5,6]. These recommendations are given to people with neurological diseases to ensure they consume a high-quality diet and achieve optimal dietary intakes to prevent malnutrition, which may help to manage some common symptoms of neurological diseases, including weight loss or gain and constipation [5,6]. Furthermore, adherence to national dietary guidelines has been shown to reduce the risk of diet-related noncommunicable diseases, including cardiovascular disease, type 2 diabetes, and obesity [5,6], which are common comorbidities of neurological diseases [7,8,9,10]. Epidemiological evidence has also indicated that type 2 diabetes, obesity, and vascular risk factors such as hypertension and dyslipidemia, are potentially modifiable risk factors for Alzheimer’s disease and dementia [11]. As the majority of the general population does not achieve the food group recommendations within national dietary guidelines [12,13,14,15], there is an opportunity for people newly diagnosed with any neurological disease to focus on improving dietary intakes in ways that are tailored to their symptoms, to improve vascular and brain health and prevent comorbidities.

Nutrition education involves a set of educational strategies to improve nutrition-related behaviors and dietary intakes that are beneficial for health and wellbeing [16]. The World Health Organization recommends nutrition education programs as a way of promoting healthy diets that are in line with national dietary guidelines to reduce the risk of noncommunicable diseases [17]. Nutrition education programs have been shown to improve nutrition-related knowledge and behaviors in the general population [18], and programs that are tailored to an individual’s dietary needs appear to be more promising at improving diet quality than non-tailored programs [19]. Best practice principles for nutrition education programs fall within five domains: (1) program design (including content areas, evidence based, goal setting, appropriate for audience including increasing recognition for co-designed programs [20], and theoretical basis); (2) program delivery (including experiential activities and fidelity); (3) educator characteristics (including expertise in content and teaching methods); (4) educator training (including initial and ongoing training); and (5) evaluation (including formative, process, and outcome evaluations, and sustained behavior change) [21].

Disease-specific programs could provide participants with messages that are tailored to their situation and enable them to share their experiences with peers who can empathize to build social support. For such programs to be effective at prompting behavior change, they must be grounded in evidence-based research and incorporate appropriate theories [22,23] and behavior change techniques (BCTs) [24]. To develop effective nutrition education programs, it is important to identify which theories and BCTs have the most potential to support the desired changes in dietary behaviors in the target population [25]. To date, it is not clear what nutrition programs exist for adults with neurological diseases, or the characteristics of such programs, including theories and BCTs.

Scoping reviews aim to map the current evidence on a topic when an area of research has not been comprehensively reviewed. Unlike systematic reviews, the purpose of this review was not to assess the effectiveness of the retrieved studies [26]. The aim of this scoping review was to explore what nutrition education programs have been implemented for adults with neurological diseases. The objectives of this review are to determine: (1) Which neurological disease populations nutrition education programs have been implemented in; (2) the characteristics of nutrition education programs; and (3) which behavior change theories and techniques have been used in the programs. A preliminary search of PROSPERO, MEDLINE, the Cochrane Database of Systematic Reviews, and JBI Evidence Synthesis revealed that there were no current or underway systematic or scoping reviews on this topic.

## 2. Methods

This scoping review was carried out according to an *a priori* protocol [27], in accordance with the Joanna Briggs Institute (JBI) methodology for scoping reviews [28] and the Preferred Reporting Items for Systematic Reviews and Meta-Analyses extension for Scoping Reviews [PRISMA-ScR] [29] (Appendix A
**Appendix A)**.

### 2.1. Inclusion Criteria

#### 2.1.1. Participants

This review considered studies that included adults (≥18 years) with any of the following neurological diseases: dementia (including Alzheimer’s disease), epilepsy, Huntington’s disease, motor neurone disease, multiple sclerosis, Parkinson’s disease, and stroke. Exclusions were: Mixed-disease populations where data for the neurological disease/s of interest could not be extracted; adults requiring medical nutrition therapy such as percutaneous gastrostomy or nasogastric tubes; and ketogenic dietary therapy for epilepsy.

#### 2.1.2. Concept

The concept considered in this review was nutrition education programs, i.e., education strategies to improve nutrition-related behaviors. Programs could be in any format (group or individual; in-person, web-based, or teleconference), of any duration (set duration or self-paced), and run for any number of sessions (single or multiple sessions). Studies reporting any of the following outcomes were considered: dietary behaviors, attitudes, or knowledge; diet quality; dietary patterns; biomarker data for nutrient or food intake; or change in intake of nutrients, energy, or food groups. We excluded: (1) dietary clinical trials with a focus on therapy or treatment and with no education component (e.g., vitamin supplementation trials); and (2) lifestyle interventions where <50% of the program pertained to nutrition.

#### 2.1.3. Context

This review considered studies that implemented a nutrition education program in any setting, including educational institutions, community centers, hospitals, care facilities, and home settings.

#### 2.1.4. Types of Studies

Both qualitative and intervention studies, with and without comparators, were considered in this review.

### 2.2. Search Strategy

A three-stage search strategy was adopted for this review, and has been described in detail in our protocol [27]. Briefly, MEDLINE and CINAHL were initially searched to identify relevant terms, which were used to develop a full search strategy (Appendix A). We searched CINAHL, Cochrane, Emcare, MEDLINE, ProQuest, and PsycInfo for published studies. We further searched Google Scholar and ProQuest Theses and Dissertations for unpublished studies and grey literature. Finally, the reference lists of all included studies were searched for additional studies. We only considered studies published in English and we did not apply any date restrictions. The initial search was conducted in August 2019 and updated in January 2022. Relevant neurological disease organizations were identified from an internet search using Google, using the terms “[disease] site:org”, “[disease] international”, and “[disease] national”, for all included neurological diseases. All organizations were contacted by email (Appendix A).

### 2.3. Study Selection

Search results were uploaded into EndNote X9 (Clarivate Analytics, Philadelphia PA, USA). One reviewer (R.D.R.) screened all titles and abstracts. The full texts of studies were imported into the JBI System for the Unified Management, Assessment and Review of Information (JBI SUMARI) (2019, Joanna Briggs Institute, Adelaide, Australia), and screened independently by two reviewers (R.D.R. and A.B.). Disagreements were resolved through discussion.

### 2.4. Data Extraction

Data were extracted using the data extraction tool specified in the protocol [27], and included: study details (author(s), year of publication, country of origin, context, study design, fidelity/drop-outs), target population (neurological disease, age, sex, sample size, comparator group details (if applicable)), and characteristics of the education program (topics, format, duration, nutrition-related outcome measures, behavior change theories, and BCTs used-assessed against Michie et al.’s 93-item taxonomy [24]). Authors were contacted to request missing or additional data, and a follow-up request sent four weeks later, as required.

## 3. Results

### 3.1. Search Results

The search strategy retrieved 3121 articles, and 2555 articles were screened by title and abstract once duplicates were removed. Full text articles were accessed for the remaining studies, and 26 were excluded (Appendix A). We emailed 61 international neurological disease organizations from the United States, Canada, the United Kingdom, Europe, Australia, and New Zealand (8 dementia; 11 epilepsy; 7 Huntington’s disease; 6 motor neurone disease; 10 multiple sclerosis; 9 Parkinson’s disease; and 10 stroke), and received responses from 35 (Appendix A); no relevant information was retrieved. Thirteen studies were included in this review (Figure 1).

### 3.2. Study Details

Table 1 shows the characteristics of the 13 included studies. The studies involved people with dementia (six studies [31,32,33,34,35,36]), people with multiple sclerosis (four studies [37,38,39,40]), stroke survivors (two studies [41,42]), and people with Parkinson’s disease (one study [43]). The studies were conducted in the United States [39,40,41,42,43], Brazil [33], Korea [31], Sweden [32], the United Kingdom [37], Germany [38], Spain [35], and Taiwan [36], and one study was conducted in three countries (France, Italy, and Spain) [34]. Eight of the studies compared an intervention group to a comparator group/s, in either a quasi-controlled trial [32], a randomized [35] or non-randomized cluster trial [34], or a randomized controlled trial [33,36,39,41,42]. Five studies did not have a comparator group [31,37,38,40,43]. Of the eight studies with comparator groups, six enlisted treatment as usual/waitlist control groups [33,34,35,36,41,42], and three had active comparator groups [32,33,39]. One study had two comparator groups: one treatment as usual group, and one active comparator group [33]. Participants in the active comparator groups received nutritional supplements [32,33] or participated in education seminars [39]. There were no qualitive studies that met the inclusion criteria.

There was very limited reporting on adherence to the nutrition interventions (fidelity), which was only reported by two studies: 27% of participants did not attend any classes, and 53% attended five out of the six classes in one study (total n = 49) [42], and 90% of participants completed all of the scheduled calls with the telehealth coach in another study, but they did not report what percentage of participants completed the online modules prior to the coaching calls [40]. The rate of drop-out from the studies was under 15% from all studies except for Brenes [43] (46.4% drop-out; Parkinson’s disease), Salva and colleagues [35] (29.4%; dementia), and Hsiao and colleagues [36] (17.4%; dementia). The rate of drop-out was unclear in one study [34] (dementia).

### 3.3. Target Populations

There were 1623 participants in the 13 included studies: 1362 with dementia and/or caregivers of people with dementia; 111 with multiple sclerosis; 135 stroke survivors, and 15 with Parkinson’s disease. The nutrition education programs were conducted with people with the disease [31,33,37,38,39,40,41,42], people with the disease and their caregivers [33,34,35,36,43], and one program was conducted with only caregivers [32]. There were 875 participants in the intervention groups, and 748 participants in the comparator groups. The total number of participants in the studies ranged from 11 [38] to 656 [35]. The median (interquartile range, IQR) age of participants was 69 (31) years, and the range of reported mean ages was 39 [38] to 84 years [31,32]. The median (range) proportion of females was 68% (38% [42]–100% [39]); one study did not report sex [31]. Other sociodemographic data were infrequently reported: seven studies reported education levels [33,35,36,39,40,42,43]; four studies reported comorbidities [32,35,36,41]; and only three studies reported the length of time since diagnosis [33,39,43].

### 3.4. Characteristics of the Nutrition Education Programs

Most of the programs were focused solely on nutrition [32,33,34,35,36,37,38,39,43], while four programs included physical activity education alongside nutrition education [40,41,42]. Seven of the programs were delivered by dietitians [32,34,35,37,39,40,41] (one of those was alongside physicians [32], and one was alongside facilitators with degrees in health education or kinesiology [40]); one was delivered by a medical student [38]; one was delivered by occupational therapists [42]. There were no details reported regarding nutrition training for non-nutrition professionals. Four programs did not specify the credentials of the facilitators [31,33,36,43]. In eleven education programs [31,32,33,34,36,37,39,40,41,42,43], participants were given instructions on how to perform the nutrition-related behaviors, including reading and interpreting food labels, preparing healthy meals, and detecting swallowing difficulties in people with dementia. Seven studies reported that participants had the opportunity to practice the skills being taught during the education sessions [32,36,37,38,40,41,42], and only four studies stated that the participants engaged in goal-setting activities [40,41,42,43]. There was no evidence of codesign in any of the nutrition education programs. Participatory research [20] was evident in two studies, and was used to inform the program topics: one program for people with multiple sclerosis used the findings from a survey [38]; and the program for people with Parkinson’s disease used focus groups [43].

Nearly all of the education programs were delivered in-person and in a group format [32,33,34,35,36,37,38,41,42], except for one telehealth intervention [40] and one online intervention [43]. One study did not report the method of delivery or program format [31]. Of the ten studies that were conducted in a group setting, only six specified that the participants engaged in group discussions [34,36,37,38,41,42]. The online program used a discussion board to facilitate group discussions [43]. Three studies had missing information on the duration of the education sessions [33,35,39] and one program was self-paced with no expected duration reported [43]. Of the studies with complete data, the total hours of program delivery ranged from 2 [38] to 36 [41] (median, IQR: 9.0, 6.6). The shortest nutrition education session lasted for 20 min [31], and the longest lasted 12 h [32] (median, IQR: 1.0, 1.3). The number of sessions for the nutrition education programs ranged from one single session (two studies, lasting two hours [38] and 12 h [32], respectively) to 36 sessions [41] (lasting one hour) (median, IQR: 6, 8). The most common frequency of delivery was weekly (six programs [31,36,37,40,42,43]). One program was delivered three days per week [41], one program involved five sessions in month one, then one session in months two, three, and six [34] and one program was delivered fortnightly [39]. The frequency of delivery was unclear in two programs: both of these programs also did not report the duration of the sessions [33,35].

The studies included a range of outcome measures to evaluate the effectiveness of the nutrition education programs. Dietary intake was assessed in six studies [37,39,40,41,42,43] using food diaries, food frequency questionnaires, and/or 24-h recalls. Nutritional status was measured using the Mini Nutritional Assessment in five studies [31,34,35,36,43], and two studies used biomarkers of nutritional status, including serum albumin, transferrin, total protein, vitamin B12, and/or hemoglobin [32,33]. Blood lipid biomarkers of dietary intake, such as triglycerides, total cholesterol, and high-density and low-density lipoprotein cholesterols, were used in two studies [41,42]. Three studies [34,36,43] evaluated the nutrition knowledge of the participants, using the Family Caregivers Nutritional Knowledge of Dementia or a nutrition knowledge questionnaire. Only one study evaluated the program effectiveness with measures of perceived benefits [39], and only one study measured perceived novelty, importance, and impact of information [38].

#### 3.4.1. Nutrition Education Programs for People with Dementia

Of the six nutrition education programs for people with dementia, four included the caregivers of people with dementia [33,34,35,36] and one program was for the caregivers only [32]. The topics included in the programs were: malnutrition and weight loss, nutritional requirements, eating behavior problems, detecting swallowing difficulties, enriching the nutritional quality of foods, altering the consistency of foods, constipation, lack of appetite, and cooking methods. Only two studies specified that the participants practiced during the sessions the nutrition-related behaviors that were being taught [32,36].

One nutrition education program was aimed at people with dementia, as opposed to their caregivers [31]. The topics included in the programs were: the concept of health, proper eating habits, nutrition and nutrients, and the problems of hyper-nutrition and nutrient deficiency. Participants were given instructions on how to perform nutrition-related behaviors, such as reading and interpreting food labels, but it was not specified if they practiced those skills during the sessions.

#### 3.4.2. Nutrition Education Programs for People with Multiple Sclerosis

All four of the nutrition education programs were delivered to people with multiple sclerosis as opposed to caregivers. Three were in-person, group programs [37,38,39] and one was a telehealth intervention [40]. Three of the programs were focused on specific diet plans: the Action and Research for Multiple Sclerosis healthy eating plan [37], the Mediterranean diet [39], and a low glycemic-load diet [40]. The topics included in the programs were: healthy eating, reading and interpreting food labels, eating out/convenience foods, meal planning, shopping tips, and cooking at home [37,39,40]. One program also included information on common study designs used in nutrition research, popular diets for multiple sclerosis, and results from clinical trials of diet and multiple sclerosis [38]. In three of the studies, instructions on how to perform the desired nutrition-related behaviors were provided to participants [37,39,40]; two of those studies also gave participants the opportunity to practice those skills during the sessions [37,40]. In half of the nutrition education programs, participants engaged in group discussions [37,38].

#### 3.4.3. Nutrition Education Programs for Stroke Survivors

The two education programs for stroke survivors were based on physical activity and nutrition education [41,42]. The nutrition topics included: preparing healthy meals, food substitutions, meal planning, and how to read and interpret food labels. The participants engaged in goal setting activities and discussed their experiences with other participants. Both programs included visual demonstrations, and participants were given time during the sessions to practice the skills being taught.

#### 3.4.4. Nutrition Education Programs for People with Parkinson’s Disease

The one nutrition education program for people with Parkinson’s disease included the caregivers alongside people with Parkinson’s disease [43]. It was a self-paced, online program consisting of short videos and handouts for each of the weekly topics, as well as videos and handouts with recipe suggestions. The weekly topics were developed from focus group discussions with people with Parkinson’s disease and included: basic nutrition; healthy eating; Parkinson’s disease and the gut; inflammation; constipation and hydration; and the protein-Levodopa interaction. The participants engaged in goal setting activities and were encouraged to regularly revise their goals. There was an online discussion board to allow participants to share their experiences with other participants.

### 3.5. Theories and Behavior Change Techniques

Five studies reported using at least one underlying behavior change theory: both of the programs for stroke survivors [41,42] used the Transtheoretical (Stage of Change) Model and one program [42] also used the Health Belief Model and the Social Cognitive Theory; one program for people with multiple sclerosis [40] used the Health Action Process Approach; one program for people with dementia and their caregivers [36] used the Knowledge-attitude-behavior Model, Bandura’s Social Learning Theory, and the integrative model of mediators of health behavior change; and the program for people and caregivers of people with Parkinson’s disease [43] used the Self-Determination Theory. However, explanations on how the theory was applied was lacking in the Brenes study [43]. A total of 22 different BCTs were used in the 13 studies in this review (Table 2). During the data extraction stage, two BCTs from two different studies were identified by only one author. The coding of these BCTs was discussed by returning to the BCT taxonomy definitions [24] and a mutual agreement was reached on whether or not the BCT was coded. The median number of BCTs used per program was six (range 1 [33] to 11 [41,42,43]); however, some of the studies lacked comprehensive details on the contents of the programs, hence it is possible that more BCTs were used but were unable to be coded. The most commonly used BCTs were: *instruction on how to perform a behavior* (eleven studies [31,32,33,34,36,37,39,40,41,42,43]), *credible source* (nine studies [32,34,35,37,38,39,40,41,42]), *behavioral practice/rehearsal* (eight studies [32,36,37,38,40,41,42,43]), *information about health consequences* (seven studies [31,34,35,36,41,42,43]), and *social comparison* (seven studies [34,36,37,38,41,42,43]). These BCTs indicate that the majority of the programs gave the participants information on how to change their behavior in relation to dietary intake, were delivered by people with relevant expertise, included practical exercises to give the participants the opportunity to practice the skills being taught, and were a source of peer exchange by way of group discussions.

## 4. Discussion

We identified the nutrition education programs that have been implemented for adults with neurological diseases, and the characteristics of those programs. In the 28 years ranging from 1993 to 2021, we only found 13 studies that met the criteria of this review. The studies included nutrition education programs for people with dementia, people with multiple sclerosis, stroke survivors, and people with Parkinson’s disease. The nutrition topics taught in the programs were evidence-based and relevant to the target group participants, but evidence of co-design was lacking. This review has identified several characteristics of programs that reflect poor design and do not align with the best practices for nutrition education programs [21]: (1) the duration and number of sessions varied between programs and the session duration was missing from nearly a third of the studies: best practice principles state that both sufficient duration and frequency are required to achieve the desired learning outcomes [21]; (2) fidelity (percentage completion of the nutrition program sessions) was rarely reported; (3) nearly half of the nutrition education programs were not delivered by dietitians or nutritionists; (4) there was no information about the initial and ongoing training for those delivering the programs; and (5) varying evaluation measures were used which indicates that dietary behavior change was not the focus for evaluation: changes in dietary intakes were not measured in more than half of the studies. The missing data and differences in the program characteristics meant that we were unable to make recommendations for future nutrition education programs for adults with neurological diseases, although weekly delivery was the most common. To facilitate evaluation and improvement of these programs, more programs need to be developed in accordance with best practice principles [21].

No nutrition education programs for people with epilepsy, Huntington’s disease, or motor neurone disease met the inclusion criteria of this review, and only one program for people with Parkinson’s disease was included. With the exception of intractable epilepsy, for which there is evidence to support the ketogenic diet as a treatment in some cases [44], national and international organizations for these neurological diseases recommend adhering to national dietary guidelines to achieve optimal nutritional intake [45,46,47]. Malnutrition and weight loss or gain are common problems for adults with neurological diseases: achieving dietary recommendations mitigates this problem and can improve quality of life for people with Huntington’s disease, motor neurone disease, and Parkinson’s disease [45,48,49]. Given that nutrition education programs can support people in meeting the dietary guidelines and achieving nutritional adequacy [18], there is a need for dietitians and nutritionists to be actively involved in using best practice principles to develop nutrition education programs for neurological diseases, particularly for people in early diagnosis to prevent malnutrition. Ultimately this could improve patient education, dietary behaviors, and quality of life for people living with different neurological diseases.

Nutrition education programs should be based on relevant theoretical frameworks [22] for enhanced efficacy [23] and better outcomes for participants [50]. Only five (38%) of the programs in this review were based on theories, and four of those were recently published. Similarly, Plow and colleagues reported that only 24% of nutrition and weight loss interventions for adults with neurological and musculoskeletal conditions were based on a behavior change theory in their systematic review [51].

Interestingly, we found that different theories were used for each of the neurological diseases (dementia, multiple sclerosis, stroke, and Parkinson’s disease), but the two programs for stroke survivors were based on the same theory despite being published twenty years apart. Similar to our findings, a systematic review reported that a range of theories have been used in nutrition education programs, including the Transtheoretical (Stages of Change) Model, Social Learning Theory, Social Cognitive Theory, Adult Learning Theory, and the Health Belief Model [23]. Theories should be used when designing and implementing nutrition interventions [21,23]; therefore, future nutrition education programs for adults with neurological diseases should adhere to best practice principles by being driven by appropriate theories, to facilitate and support the behavior changes desired by participants.

The median number of BCTs in the included studies was six; however, due to the lack of detail provided for many of the nutrition education programs in this review, it is likely that more BCTs were used in the programs. For example, we only coded *social comparison* if it was explicitly stated that participants engaged in group discussion; we did not infer this from a group setting [24]. While almost all the programs in this review were conducted in a group setting, only half specified that participants engaged in group discussions. The most common BCTs in the included studies were: *instruction on how to perform a behavior*; *credible source; behavioral practice/rehearsal;* and *social comparison.* Future programs for adults with neurological diseases should consider using these BCTs. Our findings are supported by other reviews of nutrition education interventions: *instruction on how to perform a behavior* and *social comparison* have been used in effective nutrition interventions for adults [52]; and *instruction on how to perform a behavior* and *behavioral practice/rehearsal* have been used in effective lifestyle interventions (diet and/or physical activity) for adults with chronic kidney disease [53] and obesity [54]. Furthermore, our previous systematic review of emotional wellness programs for people with multiple sclerosis found that *behavioral practice/rehearsal* was the most commonly used BCT in efficacious interventions [55]. Since it is important to identify the combination of BCTs that supports health-related behaviors [25], reporting of nutrition education programs should be in sufficient detail to allow BCTs to be identified. This would enable researchers to establish a list of BCTs that are effective for adults with neurological diseases and could be used to inform the future development of nutrition education programs.

This review was conducted following the recommendations outlined in the JBI guidelines for conducting scoping reviews, and the PRISMA-ScR checklist for scoping reviews. The search strategy was developed in consultation with a Health Sciences librarian and included published and unpublished literature. This review has some limitations. First, we only included studies in the English language. Second, a limitation of scoping reviews is that they do not include a critical appraisal of the quality of included studies or evaluate efficacy; however, the aim was to map out what programs exist, and not assess the quality or efficacy of the studies.

## 5. Conclusions

Published evidence of nutrition education programs for adults with neurological diseases is lacking, and those that are published either do not meet best practice guidelines for nutrition education programs, and/or have inconsistent characteristics. Given the role that optimal nutritional intake can play in these diseases, there is a need for dietitians and nutritionists to be involved in designing and implementing nutrition education programs that adhere to best practice guidelines, using codesign to ensure the participants’ needs are met. Such programs may help to improve patient education and dietary behaviors, therefore reducing the risk of malnutrition and comorbid diseases, which may improve quality of life. Specifically, the reporting of such programs should include the underlying behavior change theories and be in sufficient detail to allow all BCTs to be identified, including those that we have identified as most commonly used in this review. This would enable future programs to be based on appropriate theories and BCTs that appear to be effective for this population. The nutrition topics that were taught in the programs in this review were appropriate and relevant to the target group participants, and weekly delivery was most common. These characteristics should be considered when developing future nutrition education programs for adults with neurological diseases.

## Figures and Tables

**Figure 1 nutrients-14-01577-f001:**
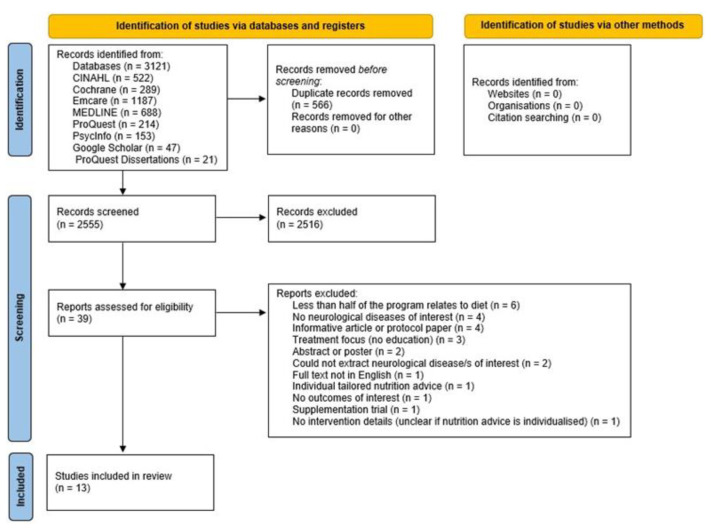
Flowchart showing the scoping review searching and screening processes [30].

**Table 1 nutrients-14-01577-t001:** Characteristics of the 13 studies that met the inclusion criteria of a scoping review of nutrition education programs for adults with neurological diseases.

Year	Author	Study Design	Sample Size (n)	AgeMean (SD) (Years)	Intervention Description	Delivery Method	Intervention Duration and Frequency	Comparator	Behavior Change Theory Used	Number of BCTs Used	Diet/Nutrition Outcome (Tool)
	**Dementia**										
2019	Cho and colleagues [31]	Pre-post	23	83.5 (4.9)	Physical activity and nutrition education for people with mild dementia. Nutrition topics: the concept of health, proper eating habits, nutrition and nutrients, and the problems of hyper-nutrition and nutrient deficiency.	NR	20 min; 16 sessions over 16 weeks	None	NR	3	Nutritional status (Mini Nutritional Assessment)
2002	Faxen-Irving and colleagues [32]	Quasi-controlled trial	33 (IG 21; CG 12)	84.0 (4.0)	Nutrition education for caregivers, plus nutritional supplements for people with dementia for 6 months. Education included practical exercises. Topics: malnutrition, food and nutritional requirements, dental care, detecting swallowing difficulties, altering food consistency.	Group, in-person	12 h; 1 session	Nutritional supplement only	NR	4	Nutritional status (serum albumin, transferrin, B12, and hemoglobin)
2020	Hsaio and colleagues [36]	RCT	57(IG 30; CG27)	74.0 (10.2)	Nutrition education for people with dementia and their caregivers, including practical exercises and demonstrations. Topics: altered eating, nutritional imbalances, Mediterranean diet preparing food, healthy fast food., videos.	Group, in-person	1 h plus 10–15 min phone calls; 6 sessions plus 3 phone calls over 3 months	Treatment as usual plus telephone counselling	Knowledge-attitude-behavior Model, Bandura’s Social Learning Theory, and the integrative model of mediators of health behavior change	6	Caregiver’s nutritional knowledge (Family Caregivers Nutritional Knowledge of Dementia); caregiver’s healthy eating behavior (Family Caregiver’s Healthy Eating Behavior for Dementia Checklist); and nutritional status (Mini Nutritional Assessment)
2011	Pivi and colleagues [33]	RCT	78(IG 25; CG1 27; CG2 26)	75.2 (76 *)	Nutrition education for people with dementia and their caregivers. Topics: nutrition in disease, behavioral changes during meals, attractive meals, constipation, hydration, administration of drugs, swallowing, food supplementation, lack of appetite.	Group, in-person	NR; 10 sessions over 6 months	CG1: treatment as usualCG2: nutritional supplement twice daily	NR	1	Nutritional status (total protein and serum albumin)
2001	Riviere and colleagues [34]	Non-randomized cluster trial	225(IG 151; CG 74)	76.3 (8.0)	Nutrition education for caregivers of people with dementia at a day hospital. Topics: weight loss consequences, eating behavior disorders, enriching food, nutritional recommendations, increasing protein and energy intake.	Group, in-person	1 h; 9 sessions over 1 year	Treatment as usual (patients and caregivers from day hospitals in France and Spain)	NR	9	Nutritional status (Mini Nutritional Assessment); and caregiver’s nutritional knowledge (Family Caregivers Nutritional Knowledge of Dementia)
2011	Salva and colleagues [35]	Cluster randomized trial	946(IG 448; CG 498)	79 (7.3)	NutriAlz nutrition program for families and caregivers of people with dementia. Topics: weight loss, nutritional monitoring, the food pyramid, menu creation, cooking methods, food substitution, eating behavior problems.	Group, in-person	NR; 4 sessions over 1 year	Treatment as usual (five patient day care centers)	NR	4	Nutritional status (Mini Nutritional Assessment)
	**Multiple sclerosis**										
1993	Doidge and colleagues [37]	Pre-post	48	46.9 (9.9)	Nutrition education for people with multiple sclerosis. Topics: *The Action and Research for Multiple Sclerosis* healthy eating plan, saturated and polyunsaturated fat, preparing food at home, understanding food labels, suitable convenience food, vitamins and minerals, weight maintenance, recipe tasting.	Group, in-person	90 min; 8 sessions over 8 weeks	None	NR	8	Diet composition (daily energy intake and nutrient intakes)
2019	Katz Sand and colleagues [39]	Pilot RCT	34(IG 18; CG 16)	43 (NR)	Nutrition education for people with multiple sclerosis (groups of five); Mediterranean Diet. Topics: shopping tips, sample menu plan, reading food labels, eating at restaurants. Participants returned monthly (or dialed in) to discuss issues with following the diet.	Group, in-person and/or telehealth	NR; 6 sessions over 6 months	MS education seminars	NR	6	Dietary adherence and food group intake (food frequency questionnaire); and perceived benefits
2016	Riemann-Lorenz and colleagues [38]	Single aim, post	11	38.5 (12.3)	Nutrition education for people with multiple sclerosis (1 session), including 2 short group discussions. Topics: epidemiology, research study designs, study endpoints and problems, experiences with multiple sclerosis diets, common multiple sclerosis diets, RCTs of diet and multiple sclerosis.	Group, in-person	2 h; 1 session	None	NR	3	Novelty of information/knowledge; importance of information; and impact of information
2020	Wingo and colleagues [40]	Single arm, post	18	46.0 (11.6)	Nutrition education and physical activity education for people with multiple sclerosis, for the low glycemic index diet, including online modules and calls from tele-coaches. Nutrition topics: meal planning, foods to eat and limit, cooking basics, healthy eating on a budget. Weeks 1–5 were standardized information. Weeks 6–12 were tailored to address barriers and goals.	Individual, telehealth	12 online modules (time NR) and 12 20–45 min phone calls over 12 weeks	None	Health Action Process Approach	10	Diet quality (24-h food recall); and fat mass (dual-energy X-ray absorptiometry scan)
	**Stroke**										
2000	Rimmer and colleagues [41]	RCT	35(IG 18; CG 17)	53.2 (8.3)	Health Promotion program for stroke survivors (exercise, nutrition, and health behavior classes), including cooking demonstration and practice. Nutrition topics: low-fat and low-cholesterol foods, preparation of healthy meals, healthy food substitutes.	Group, in-person	1 h; 36 sessions over 12 weeks	Waitlist controls	Transtheoretical (Stage of Change) Model	11	Dietary fat intake (Rate Your Plate Eating Pattern Assessment) and blood lipid profile (total cholesterol, high-density and low-density lipoprotein cholesterols, triglycerides)
2020	Towfighi and colleagues [42]	RCT	100(IG 49; CG 51)	58.0 (9.0)	*Healthy Eating and Lifestyle After Stroke* program for stroke survivors. Nutrition topics: healthy dietary patterns, monitoring food intake, food label reading, shopping, purchasing healthy foods, diet as a means of secondary stroke prevention.	Group, in-person	2 h; 6 sessions over 6 weeks	Treatment as usual	Transtheoretical (Stage of Change) Model, Health Belief Model, and Social Cognitive Theory	11	Serves of fruits/vegetables per day; waist circumference; and blood lipid profile (total cholesterol, high-density and low-density lipoprotein cholesterols, triglycerides, hemoglobin A1c)
	**Parkinson’s disease**										
2000	Brenes [43]	Pre-post	15	69.0 (NR)	Virtual nutrition education program for people with Parkinson’s disease and their caregivers. Included lesson videos, handouts and recipes (video and written). Topics: basic nutrition, healthy eating, Parkinson’s disease and the gut, inflammation and Parkinson’s disease, constipation and hydration, and ‘protein and Levodopa.	Individual, online	Self-paced; 6 sessions over 6 weeks	None	Self-Determination Theory	11	Nutritional status (Mini Nutritional Assessment); intake of macronutrients, micronutrients, and food groups (Diet History Questionnaire 3); nutrition knowledge (nutrition knowledge questionnaire); motivation about nutrition knowledge

SD, standard deviation BCTs, behavior change techniques; NR, not reported; IG, intervention group; CG, comparator group; RCT, randomized controlled trial. * Median reported.

**Table 2 nutrients-14-01577-t002:** Behavior change techniques used in each of the 13 studies that met the inclusion criteria of a scoping review of nutrition education programs for adults with neurological diseases.

	Cho and Colleagues [31]	Doidge and Colleagues [37]	Faxen-Irving and Colleagues [32]	Hsiao and Colleagues [36]	Katz Sand and Colleagues [39]	Pivi and Colleagues [33]	Riemann-Lorenz and Colleagues [38]	Rimmer and Colleagues [41]	Riviere and Colleagues [34]	Salva and Colleagues [35]	Towfighi and Colleagues [42]	Wingo and Colleagues [40]	Brenes [43]	Total n
Instruction how to perform a behavior														**11**
Credible source														**9**
Behavioral practice/rehearsal														**8**
Information about health consequences														**7**
Social comparison														**7**
Self-monitoring behavior														**5**
Demonstration of the behavior														**5**
Problem solving														**4**
Adding objects to the environment														**4**
Social support (unspecified)														**3**
Goal setting (outcome)														**3**
Framing/reframing														**3**
Feedback on behavior														**3**
Action planning														**2**
Reduce negative emotions														**2**
Prompts/cues														**2**
Review behavior goal(s)														**2**
Monitoring behavior by others without feedback														**1**
Monitoring outcome(s) by others without feedback														**1**
Biofeedback														**1**
Social support (practical)														**1**
Goal setting (behavior)														**1**
**Total n**	**3**	**8**	**4**	**6**	**6**	**1**	**3**	**11**	**9**	**4**	**11**	**10**	**11**	

## Data Availability

Not applicable.

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
