# Peer review of "Nutrition Education Programs for Adults with Neurological Diseases Are Lacking: A Scoping Review"

_nutrients, 2022, doi:10.3390/nu14081577_

Round 1

Reviewer 1 Report

The authors evaluated the existing data on nutrition education programs for adults with neurological diseases.The aim of this scoping review was to map the current evidence on what nutrition education programs have been implemented for adults with neurological diseases, an area of research that has not been comprehensively reviewed.

The authors did define a clear aim in the introduction, however the aim is not clearly underlined in the Abstract.

In paragraph 2.3. line 136-137 – “Disagreements were resolved through discussion.” Authors were not clear on what were considered disagreements and what the resolution parameters were.

Table 5 headings are not clearly visible due to overlapping of writing.

References cited were not consistent in citation style throughout the text, e.g. Citations in paragraph 3.3. (…the length of time since diagnosis.[33, 39, 43]) vs paragraph 3.4 (…programs also did not report the duration of the sessions [33, 35].)

Author Response

Reviewer comment: The authors evaluated the existing data on nutrition education programs for adults with neurological diseases. The aim of this scoping review was to map the current evidence on what nutrition education programs have been implemented for adults with neurological diseases, an area of research that has not been comprehensively reviewed.

Author response: We thank the reviewer for their comments, which have been addressed below and in the manuscript.

Reviewer comment #1:

The authors did define a clear aim in the introduction, however the aim is not clearly underlined in the Abstract.

Author response #1:

We have added the aim into the abstract (new text underlined below).

Lines 11-13: “Nutrition education programs can support people in adhering to guidelines; hence the aim of this scoping review was to explore what programs have been implemented for adults with neurological diseases.”

Reviewer comment #2:

In paragraph 2.3. line 136-137 – “Disagreements were resolved through discussion.” Authors were not clear on what were considered disagreements and what the resolution parameters were.

Author response #2:

We have included details in the results outlining what disagreements arose and how they were resolved (new text underlined below).

Lines 306-309: “During the data extraction stage, two BCTs from two different studies were identified by only one author. The coding of these BCTs was discussed by returning to the BCT taxonomy definitions [24] and a mutual agreement was reached on whether or not the BCT was coded.”

Reviewer comment #3:

Table 5 headings are not clearly visible due to overlapping of writing.

Author response #3:

We have corrected the table number (Table 2, page 14), and heightened rows as required so that all headings are visible.

Reviewer comment #4:

References cited were not consistent in citation style throughout the text, e.g. Citations in paragraph 3.3. (…the length of time since diagnosis.[33, 39, 43]) vs paragraph 3.4 (…programs also did not report the duration of the sessions [33, 35].)

Author response #4:

All citations have been checked and corrected where necessary to be consistent throughout the text.

Reviewer 2 Report

Authors wrote a complete review of the literature.

I would suggest to clearly define the aim and hypothesis of the Manuscript. 

Also I highly suggest to deeply discuss why these different programs are important for the patient. 

Authors could also discuss how to improve patient education

Author Response

Reviewer comment: Authors wrote a complete review of the literature.

Author response: We thank the reviewer for their comments, which have been addressed below and in the manuscript.

Reviewer comment #1:

I would suggest to clearly define the aim and hypothesis of the Manuscript.

Author response #1:

We have added the aim into the abstract, consistent with the aim defined in paragraph 4 of the introduction (new text underlined below).

Lines 11-13: “Nutrition education programs can support people in adhering to guidelines; hence, the aim of this scoping review was to explore what programs have been implemented for adults with neurological diseases.”

Reviewer comment #2:

Also I highly suggest to deeply discuss why these different programs are important for the patient.

Author response #2:

We have discussed the importance of nutrition education programs for adults with neurological diseases in the introduction and the discussion of the manuscript. We began by outlining some of the benefits from achieving a high-quality diet (prevention of malnutrition, reduced risk of common comorbidities, including cardiovascular disease, type 2 diabetes, and obesity, and improved vascular and brain health; introduction, lines 39-52). We then described the role that nutrition education programs play in helping people achieve a high-quality diet in line with dietary recommendations (improving their nutrition-related knowledge and dietary behaviors; introduction, lines 53-58), and why disease-specific programs are beneficial (tailored messages and peer-sharing; introduction, lines 78-70). In the discussion we also outlined how achieving the dietary recommendations could benefit people with neurological diseases (mitigate common symptoms such as weight loss or gain and improved quality of life; discussion, lines 352-362).

Reviewer comment #3:

Authors could also discuss how to improve patient education.

Author response #3:

We have revised part of the discussion and conclusion to highlight how our review findings could contribute to improve patient education (new text underlined below).

Lines 355-358: “…dietitians and nutritionists to be actively involved in using best practice principles to develop nutrition education programs for neurological diseases, particularly for people in early diagnosis to prevent malnutrition. Ultimately this could improve patient education, dietary behaviors, and quality of life for people living with different neurological diseases.”

Lines 413-471: “…there is a need for dietitians and nutritionists to be involved in designing and implementing nutrition education programs that adhere to best practice guidelines, using codesign to ensure the participants’ needs are met. Such programs may help to improve patient education and dietary behaviors, therefore reducing the risk of malnutrition and comorbid diseases, which may improve quality of life.”